# How ligands regulate the binding of PARP1 with DNA: Deciphering the mechanism at the molecular level

**Kai Wang** [1]*, **Yizhou Wu**[1], **Lizhu Lai**[1], **Xin Wang**[1], **Shuya Sun**[2]*

**1** School of Agriculture and Biology, Zhongkai University of Agriculture and Engineering, Guangzhou, P. R. China, **2** School of Traditional Chinese Medicine, Guangzhou University of Chinese Medicine, Guangzhou, P. R. China

* wangkai@zhku.edu.cn (KW); sunshy@gzucm.edu.cn (SS)

**Data Availability Statement:** All relevant data are within the paper and its Supporting Information files.

**Funding:** This research was funded by the National Natural Science Foundation of China, grant number

## Abstract

The catalytic (CAT) domain is a key region of poly (ADP-ribose) polymerase 1 (PARP1), which has crucial interactions with inhibitors, DNA, and other domains of PARP1. To facilitate the development of potential inhibitors of PARP1, it is of great significance to clarify the differences in structural dynamics and key residues between CAT/inhibitors and DNA/PARP1/inhibitors through structure-based computational design. In this paper, conformational changes in PAPR1 and differences in key residue interactions induced by inhibitors were revealed at the molecular level by comparative molecular dynamics (MD) simulations and energy decomposition. On one hand, PARP1 inhibitors indirectly change some residues of the CAT domain which interact with DNA and other domains. Furthermore, the interaction between ligands and catalytic binding sites can be transferred to the DNA recognition domain of PARP1 by a strong negative correlation movement among multi-domains of PARP1. On the other hand, it is not reliable to use the binding energy of CAT/ligand as a measure of ligand activity, because it may in some cases differs greatly from the that of PARP1/DNA/ligand. For PARP1/DNA/ligand, the stronger the binding stability between the ligand and PARP1, the stronger the binding stability between PARP1 and DNA. The findings of this work can guide further novel inhibitor design and the structural modification of PARP1 through structure-based computational design.

## Introduction

PARP1 (poly (ADP-ribose) polymerase 1), a complex protease with multiple domains, has been widely investigated as a popular target for many diseases such as cancers [1–7]. Each domain of PARP1 provides a significant structural basis for realizing multiple functions of PARP1 [8–13]. The main function of the N-terminal DNA-binding domain (DBD) is to recognize and repair damaged DNA structures under certain conditions. The recognition and repair mechanisms of the DBD have been preliminarily elucidated, including the functional mechanism of zinc ions in the whole recognition process [14–16]. The BRCT (breast cancer

21803079. The funders had no role in study design, data collection and analysis, decision to publish, or preparation of the manuscript.

**Competing interests:** The authors have declared that no competing interests exist.

susceptibility gene (BRCA) C-terminus) domain can mediate the DNA transfer of PARP1 [9,13]. Furthermore, some studies have focused on developing potential selective PARP1 inhibitors targeted to BRCT [17]. The WGR (tryptophan-glycine-arginine-rich) domain, with the main sites of self-modification, shows interaction with other domains and DNA in some crystal structures [8]. The C-terminal catalytic (CAT) domain contains binding sites for NAD + and catalytic sites for the synthesis of PAR-(poly(ADP-ribose)). In addition, the CAT domain contains a regulatory helical subdomain (HD) and ADP-ribosyl transferase (ART), which provides a significant structural basis for the structure-based de novo design of PARP1 inhibitors [18,19]. Although many scientific issues have been addressed and significant milestones have been reached, it is still very challenging to elucidate the functional mechanism of PARP1 and to develop PARP1 inhibitors [2,4,18,20].

MD simulation has been broadly employed to unveil the detailed interaction mechanisms between drugs and target proteins [14,21,22]. Inhibitors related to the CAT domain of PARP1 have been widely studied through experiments and computational methods [2,12,19,23]. However, how the inhibitor molecules affect the conformational and functional changes in PARP1 bound to DNA is still unclear. Knowledge on the difference in interaction between PARP1/ inhibitors and PARP1/DNA/inhibitors can assist in improving the accuracy of virtual screening or structure-based de novo inhibitor design, thereby accelerating the speed of the inhibitor design of PARP1.

In the present study, the binding mechanism changes induced by PARP1 inhibitors were investigated by MD simulation. The energy decomposition method was used to analyze the difference in the energy contribution of each residue of CAT between CAT/inhibitors and DNA/PARP1/inhibitors. The kinetic correlation analysis of each residue was used to distinguish the dynamic changes brought by the inhibitors. By clarifying the differences in the interaction mechanisms between CAT and CAT/DNA induced by inhibitors with different activities, our research will guide further novel inhibitor design and the structural modification of PARP1 through structure-based computational design, and assist in speeding up the progress of cancer treatment.

## Materials and methods

### Construction of initial models

DSB/CAT/ZnF1/ZnF3/ligand crystal structures were downloaded from the Protein Data Bank [24,25]. Eight systems were constructed to examine the conformational changes and the binding mechanism of PARP1, as shown in **Table 1**. To analyze the interaction between the WGR/ CAT and DBD domain, the CAT domain defined here includes the WGR structure, which means that the current CAT sequence consists of three parts: the residues from 1 to 107

Table 1. Constructed models.

| Name | System |
|---|---|
| Model 1 | DSB/ZnF1/ZnF3/CAT/ |
| Model 2 | DSB/ZnF1/ZnF3/CAT/Lig1 |
| Model 3 | DSB/ZnF1/ZnF3/CAT/Lig2 |
| Model 4 | DSB/ZnF1/ZnF3/CAT/Lig3 |
| Model 5 | CAT |
| Model 6 | CAT/Lig1 |
| Model 7 | CAT/Lig2 |
| Model 8 | CAT/Lig3 |

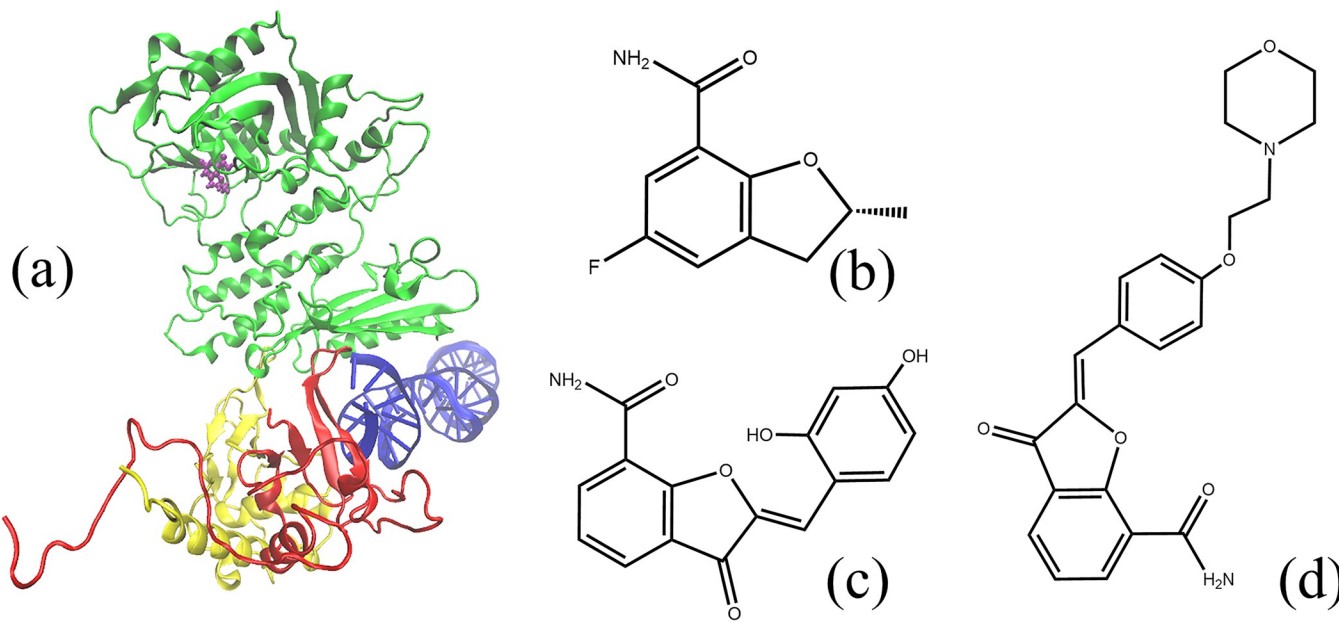

**Fig 1.** (a) Three-dimensional diagram of the binding system (4OPX). ZnF1, ZnF3 and CAT (including WGR) are depicted in red, yellow, and green, respectively. DSB is depicted in blue. Ligand is shown in purple stick model. (b) Two-dimensional diagram of $C_{10}H_{10}FNO_2$, marked as Lig1. (c) Two-dimensional diagram of $C_{16}H_{11}NO_5$, marked as Lig2. (c) Two-dimensional diagram of $C_{22}H_{22}N_2O_5$, marked as Lig3.

(WGR), sequence 131 to 250 (HD) and residue 257 to 483 (ART). Structural optimization and the protonation process of zinc ions were performed with MOE software. All hydrogen atoms were modified for each model by the leap module of Amber20 [26]. Subsequently, the TIP3P water environment with the boundary extension of a 10Å cubic water box was constructed for each model. Chlorine and sodium ions were added to neutralize the charge of all models.

A crystal structure of PARP1 with double-strand breaks (DSB) was used to investigate the conformational changes induced by the ligand [27]. **Fig 1A** shows the overall three-dimensional diagram of the system. (**Fig 1B–1D**) shows the structure of three ligands, marked as Lig1, Lig2 and Lig3. The names of Lig1, Lig2 and Lig3 are (2R)-5-fluoro-2-methyl-2,3-dihydro-1-benzofuran-7-carboxamide, (2Z)-2-(2,4-dihydroxybenzylidene)-3-oxo-2,3-dihydro-1-benzofuran-7-carboxamide, and (2Z)-2-{4-[2-(morpholin-4-yl)ethoxy]benzylidene}-3-oxo-2,3-dihydro-1-benzofuran-7-carboxamide, respectively. These three ligands all contain nicotinamide groups, which are active groups contained in a common PARP inhibitor. In addition, the size of these three ligands is quite different, which is suitable for further dynamics studies. The IC50 values of binding affinity are 2450 (nM) for Lig1, 753 (nM) for Lig2, and 2070 (nM) for Lig3, respectively. Eight systems were constructed to describe the interactions among the inhibitors, the multi-domains of PARP1, and the DSB of DNA, as shown in **Table 1**. The original PDB code for these models contains 4OPX, 4OQA and 4OQB, but not ZnF2 or BRCT. In this system, ZnF1, ZnF3 and other domains are disconnected. Model 1 was obtained by removing the ligand structure of 4OPX, and Models 2, 3, and 4 were derived from 4OPX, 4OQA, and 4OQB, respectively. Model 5 was obtained by extracting the CAT structure in model 1. Models 6, 7, and 8 were obtained by extracting the ligands and CAT structures from Models 2, 3, and 4. To describe the changes in binding energy among these domains accurately, the original structure was used in MD simulations without any modifications. More importantly, WGR and CAT are linked together in these systems. Therefore, in this paper, we classify the WGR structure as the CAT domain, which is uniformly called CAT, to facilitate

the calculation of the impact of inhibitors on PARP1 structure and the binding stability of PARP1 to DSB. Currently, the CAT structure includes three parts: WGR (residue 1 to 107; the corresponding serial number in the PDB sequence is residue 531 to 537), HD (sequence 131 to 250; the corresponding serial number in the PDB sequence is residue 661 to 880) and ART (sequence 257 to 481; the corresponding serial number in the PDB sequence is residue 787 to 1011). In the following part, we will analyze the motion correlation between these three subdomains.

For the selection of the initial structure, the following description was required. Although the protein database contains dozens of crystal structures (including PARP1 and its inhibitors), 4OPX, 4OQA and 4OQB are the only crystal structures that contain PARP1, ligands and DSB. Although the complex system between DSB, PARP1 and its inhibitors can be predicted by homologous modeling and molecular docking, the structures predicted by computing methods needs to be verified by experimental methods or quantitative methods. These structures will not only introduce some errors in the binding mode between the inhibitors and PARP1, but also require more experimental and computational research to further study the regulatory mechanisms of ligands in PARP1 and DSB. Therefore, these three structures were used in this work.

## MD simulation

Amber20 with GPU acceleration was used to perform all MD simulations. The Amber FF19SB and gaff2 force fields were parameterized for proteins and ligands [28]. OL15 force fields were used to perform the dynamics of DNA [29]. The cut-off distance for van der Waals and electrostatic non-bonded interactions was set as 12 Å. The zinc parameters and distance restraints with 1000 kcal/(mol·Å$^2$) were applied for Zn1 in ZnF1 and Zn3 in ZnF3 to keep their 4-fold coordination features. Gauss was used to optimize the structure and calculate the charge parameters of the ligands. Furthermore, the method of particle mesh Ewald (PME) was used to calculate the long-range electrostatic interactions [30], and the SHAKE algorithm was used for covalent bonds with hydrogen atoms. Multiple steps were carried out to optimize and stabilize the initial systems. Firstly, all water molecules were optimized by restricting proteins, DNA, salt ions, ligand molecules and zinc ions. Secondly, the side chains of the proteins and DNA were optimized with a water environment. Thirdly, all restrictive conditions except the restraints of zinc ions were released to achieve the dynamic movement of molecules such as proteins and ligands. Additionally, the system was first heated from 0 K to 310 K in 50 ps using Langevin dynamics at a constant volume, and then equilibrated for 400 ps at a constant pressure of 1 atm. A weak constraint of 10 kcal/(mol·Å2) was used to restrain all the heavy atoms during the heating process. Periodic boundary dynamic simulations were carried out for all systems with an NPT ensemble (constant composition, pressure, and temperature). Trajectories obtained from MD simulations of 500 ns were sampled for Model 1 to Model 4 to describe the interactions among the ligands, PARP1 and DSB. For complex Models 5 to 8, MD simulations of 100 ns were performed to calculate the binding energies and structural dynamics of these ligands. **S1 Fig in S1 File** shows the variation in RMSD values of each structure with the time of MD simulation, indicating that all systems have reached equilibrium. Therefore, conformational analysis and kinematic correlation analysis can be performed based on these structure samplings. Finally, 10 ns simulation was carried out to equilibrate the motion of all molecules, and then the long-scale MD simulation was performed. The trajectories were analyzed using Cpptraj and VMD [31,32]. The binding free energy of the complexes and the energy decomposition of the per-residue contributions were calculated using the MMGBSA method [33]. It is worth mentioning that the last 20 ns of the MD simulation trajectory was

used to calculate the binding free energy to avoid errors caused by instability in the early stage of the MD simulation.

## Calculation of linear correlations

Linear correlation coefficients [34–37] between each two residues of CAT were calculated as follows:

$$R(x, y) = \frac{Cov(x, y)}{stdev(x) * stdev(y)} \tag{1}$$

The variables x and y represent the value obtained by subtracting the average position from the centroid position of any residue. Cov(x, y) is the covariance between x and y. The variables stdev(x) and stdev(y) are the standard deviations of x and y. All snapshots of MD trajectories need to be matched based on the first snapshot to eliminate the errors caused by the translation and rotation of residues in MD simulations.

# Results and discussion

## Conformational changes in ZnF1

Previous studies have clarified the functional mechanism of the DBD binding to damaged DNA [10,11,14–16,38,39]. The main function of ZnF1 and ZnF2 is to identify and stabilize DNA structures, which provides a structural basis for the further repair of damaged DNA. However, it is unclear how PARP1 inhibitors directly or indirectly affect the interaction between DBD and DNA. MD simulations were used to explore at the molecular level the conformational changes in ZnF1 binding with DSB induced by different ligands. Firstly, RMSF was used to measure the flexibility change in each residue in ZnF1, as shown in **Fig 2A**. The existence of ligands and the change in ligand structure have little effect on the structural flexibility of ZnF1. **Fig 2B** shows the residues at the interface of ZnF1 binding with DSB and CAT. Met38 and Phe39 are both involved in the binding process of ZnF1 binding with DSB and CAT. The interaction interface among ZnF1, DSB, and CAT was determined based on the distance between interacted atoms in the crystal structure of 4OPX. The specific steps are as follows: selecting a range of 4 angstroms among the atoms of residues or nucleotides, connecting these discontinuous residues together by other connected residues, and defining them as the interaction interface. Secondly, to accurately describe the ligand interaction between ZnF1-binding DSB and CAT, the MMGBSA method is used to calculate the binding energy of each complex system [33]. **Table 2** shows the binding free energies of ZnF1/DSB binding with different ligands. Among them, Lig1 and Lig2 interact with CAT and weaken the binding stability between ZnF1 and DSB. The interaction between Lig1 and CAT indirectly leads to the weakening of the van der Waals interaction and the strengthening of the electrostatic interaction between ZnF1 and DSB. Furthermore, the change in polar solvation free energy is larger than that in non-polar solvation free energy. On the other hand, the reduction in the binding energy of ZnF1/DSB generated by Lig2 is mainly reflected in the reduction in the van der Waals interaction and polar solvation free energy. For Lig1 and Lig2 binding to CAT, the binding stability between ZnF1 and DSB is indirectly weakened, while the combination of Lig3 and CAT exerts slight effect on ZnF1 binding to DSB. What's more, **Fig 2C** describes the comparation of the per-residue energy contribution spectra of ZnF1 on the surface of ZnF1/DSB induced by Lig1, Lig2 and Lig3. It shows that the ligand molecules indirectly affect the binding energy of some residues in the process of ZnF1 binding to DSB. For example, due to the influence of Lig2, the binding forces of A9, K10 and M38 in ZnF1 are significantly strengthened,

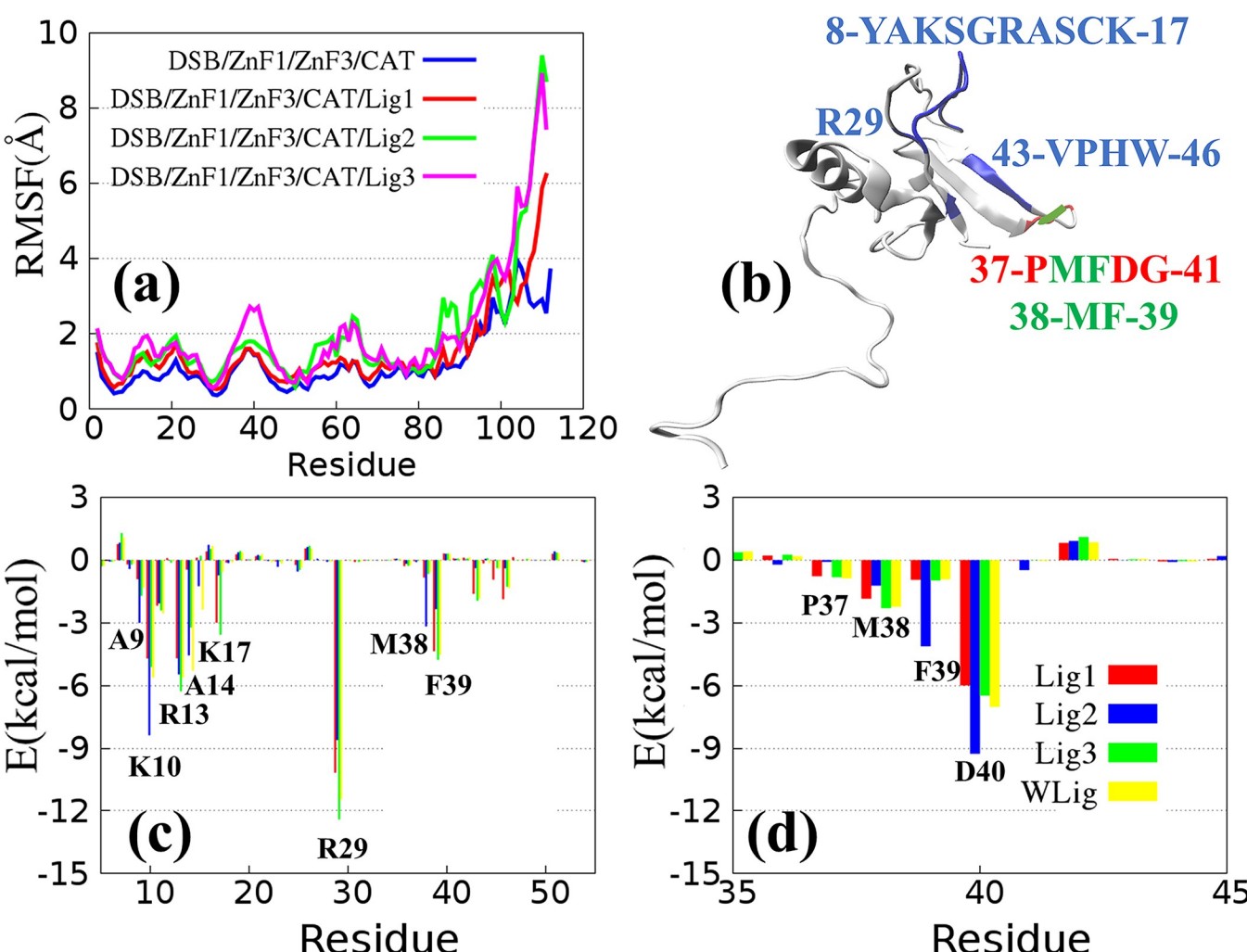

**Fig 2.** (a) Per-residue RMSFs for ZnF1 in Model 1 to Model 4. (b) The three-dimensional structure diagram of ZnF1 (4OPX). The interaction interface of ZnF1/DSB and ZnF1/CAT are represented by blue and red, respectively, and the corresponding residues are shown in blue and red. (c) Per-residue energy contribution spectra of ZnF1 on the surface of ZnF1/DSB for Model 1 to Model 4. Red, blue, green, and yellow bars represent Lig1, Lig2, Lig3, and without ligand binding to CAT (marked as WLig). (d) Per-residue energy contribution spectra of ZnF1 on the surface of ZnF1/CAT for Model 1 to Model 4. Red, blue, green, and yellow bars represent Lig1, Lig2, Lig3, and WLig.

whereas the binding forces of R29 and F39 in ZnF1 are obviously weakened. On this basis, the change in binding stability between ZnF1 and CAT was further studied to explore whether such effects are transmitted through CAT.

Table 3 shows the variations in the binding free energies of ZnF1/CAT induced by the three ligands. It can be noted that Lig2 slightly enhances the binding stability of ZnF1 with CAT, whereas Lig1 and Lig3 have little influence on ZnF1 binding to CAT. The energy decomposition of key residues of ZnF1 in Fig 2D supports this finding. Influenced by Lig2 binding to CAT, the interaction of F39 and D40 residues is strengthened for ZnF1/CAT, while other residues and ligands are slightly influenced. The binding affinity of Lig2 is better than others. Therefore, the ligand with better affinity can strengthen the binding stability between the ZnF1 and CAT domains. Additionally, this can also weaken the binding stability of ZnF1 binding to DSB, thus realizing the function of inhibiting PARP1.

**Table 2. The calculated (MMGBSA) binding free energies of the ZnF1/DSB for Model 1 to Model 4.**

| Term | ZnF1/DSB | ZnF1/DSB ~Lig1 | ZnF1/DSB ~Lig2 | ZnF1/DSB ~Lig3 |
|---|---|---|---|---|
| $\Delta E_{vdw}^a$ | -51.01 | -40.31 | -48.25 | -51.13 |
| $\Delta E_{ele}^b$ | -774.63 | -853.58 | -774.98 | -850.86 |
| $\Delta E_{polar}^c$ | 778.03 | 863.45 | 786.61 | 855.41 |
| $\Delta E_{nonpolar}^d$ | -6.14 | -5.07 | -6.10 | -6.20 |
| $\Delta G^e$ | -53.15 | -35.51 | -42.72 | -52.78 |

[a] Van der Waals energy.

[b] Electrostatic energy.

[c] Polar solvation free energy.

[d] Nonpolar solvation free energy.

[e] Calculated Gibbs free energy.

All the energy terms are in kcal/mol.

## Conformational changes in ZnF3

Previous studies have shown that the main function of ZnF3 is to connect DSB and CAT domain of PARP1 [11]. ZnF3 plays no role in the recognition and stabilization of DNA. Therefore, the conformation change and binding energy change in ZnF3 in different ligand systems were investigated. As described in **Fig 3A**, the three ligand molecules exert slight influence on the structural flexibility of ZnF3, except the case of DSB/ZnF1/ZnF3/CAT/Lig2, where the flexibility increases slightly from residue 40 to 60. Furthermore, the residues at the interface of ZnF3 binding with DSB and CAT are shown in **Fig 3B**. Unlike ZnF1, there are no overlapping residues between these two regions of ZnF3.

 **Table 4** shows the variation in the binding free energies of ZnF3/DSB induced by different ligands. The interaction between ZnF3 and DSB is relatively weak, and its value tends to be 0. However, the Lig2 to CAT binding structure tends to slightly strengthen the binding stability of ZnF3 and DSB. This is inferred because the per-residue energy contribution of ZnF3 to Lig2 is quite different from the other three cases (**Fig 3C**). Firstly, the binding free energy contribution at K38, Q47, and V48 is significantly strengthened. Secondly, the repulsive force at K56 is obviously stronger than others. Finally, the binding stability at S50 and G51 is significantly weakened. Among them, the van der Waals interaction and electrostatic interaction in the binding process of ZnF3 and DSB are both strengthened, indicating the slightly stronger attraction of ZnF3 to DSB. More importantly, these residues with large changes in energy contribution overlap with the region where RMSF flexibility slightly increases. Therefore, the increase in flexibility for residues of ZnF3 is closely related to the change in the binding free energy of ZnF3 with DSB.

**Table 3. The calculated (MM/GBSA) binding free energies of ZnF1/CAT.**

| Term | ZnF1/CAT | ZnF1/CAT ~Lig1 | ZnF1/CAT ~Lig2 | ZnF1/CAT ~Lig3 |
|---|---|---|---|---|
| $\Delta E_{vdw}$ | -30.10 | -31.10 | -43.02 | -29.92 |
| $\Delta E_{ele}$ | -84.44 | -86.50 | -97.93 | -175.16 |
| $\Delta E_{polar}$ | 91.42 | 94.93 | 115.69 | 182.89 |
| $\Delta E_{nonpolar}$ | -3.73 | -3.71 | -5.47 | -4.08 |
| $\Delta G$ | -26.85 | -26.38 | -30.73 | -26.27 |

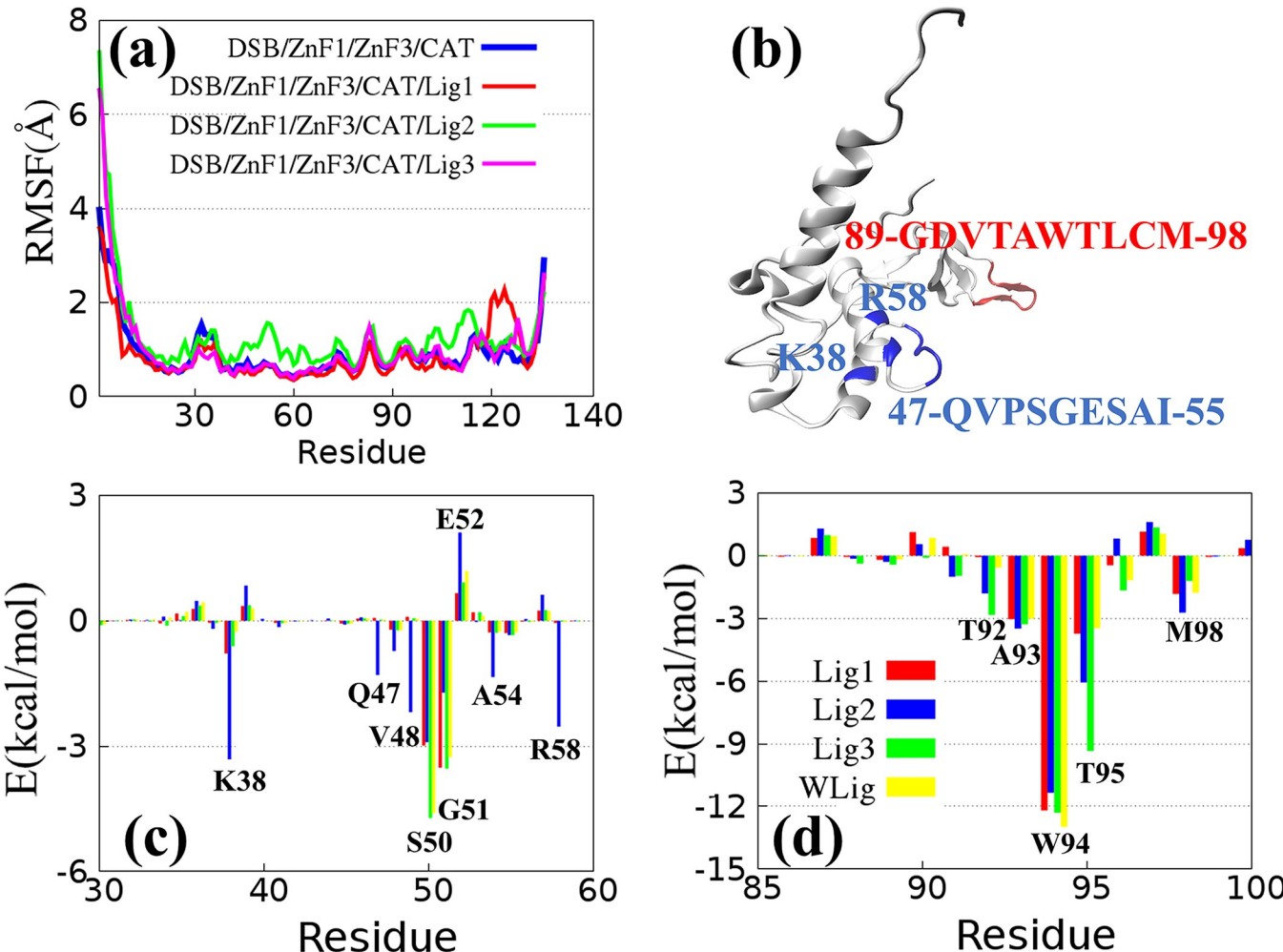

**Fig 3.** (a) Per-residue RMSFs for ZnF 3 in Model 1 to Model 4. (b) The three-dimensional structure diagram of ZnF3 (4OPX). The interaction interfaces of ZnF3/DSB and ZnF3/CAT are represented by blue and red, respectively, and the corresponding residues are shown in blue and red. (c) Per-residue energy contribution spectra of ZnF3 on the surface of ZnF3/DSB. Red, blue, green, and yellow bars represent Lig1, Lig2, Lig3, and WLig. (d) Per-residue energy contribution spectra of ZnF3 on the surface of ZnF3/CAT. Red, blue, green, and yellow bars represent Lig1, Lig2, Lig3, and WLig.

**Table 5** shows the binding free energies of ZnF3/CAT binding with different ligands. The combination of Lig3 and CAT increases the binding free energy of CAT and ZnF3, which is mainly reflected by the enhancement in energy contribution of the T92 and 95 residues (**Fig 3D**). In conclusion, although PARP1 inhibitors have a certain ability to affect the conformational changes in ZnF3, this ability is relatively weak. Furthermore, based on the conformational analysis of ZnF1 and ZnF3, the binding free energy between ZnF1 and ZnF3 is given in

**Table 4. The calculated (MMGBSA) binding free energies of ZnF3/DSB.**

| Term | ZnF3/DSB | ZnF3/DSB ~Lig1 | ZnF3/DSB ~Lig2 | ZnF3/DSB ~Lig3 |
|---|---|---|---|---|
| $\Delta E_{vdw}$ | -10.26 | -10.78 | -29.65 | -11.10 |
| $\Delta E_{ele}$ | -420.70 | -468.41 | -719.34 | -841.93 |
| $\Delta E_{polar}$ | 435.08 | 482.81 | 750.06 | 855.73 |
| $\Delta E_{nonpolar}$ | -1.43 | -1.58 | -3.62 | -1.52 |
| $\Delta G$ | 2.68 | 2.04 | -2.55 | 1.18 |

**Table 5. The calculated (MMGBSA) binding free energies of ZnF3/CAT.**

| Term | ZnF3/CAT | ZnF3/CAT ~Lig1 | ZnF3/CAT ~Lig2 | ZnF3/CAT ~Lig3 |
|---|---|---|---|---|
| $\Delta E_{vdw}$ | -51.64 | -46.63 | -49.52 | -64.60 |
| $\Delta E_{ele}$ | -148.88 | -110.54 | -84.86 | -102.16 |
| $\Delta E_{polar}$ | 179.12 | 140.09 | 117.22 | 137.43 |
| $\Delta E_{nonpolar}$ | -6.81 | -6.20 | -6.55 | -8.39 |
| $\Delta G$ | -28.20 | -23.28 | -23.70 | -37.72 |

**Table 6**. Lig1 indirectly strengthens the binding stability of ZnF1 with ZnF3 by improving the polar solvation free energy. In contrast, the binding stability of ZnF1 with ZnF3 is significantly weakened by Lig2 due to the reduction in polar solvation free energy. Due to the combination of the electrostatic interaction and polar solvation free energy, Lig3 has little influence on the binding stability of ZnF1 and ZnF3. These binding stabilities are related to the activity of PARP1 inhibitors, but not in a linear manner. To explore their relationship, it is necessary to analyze the change in the binding stability from the conformational change in CAT induced by PARP1 inhibitors.

## Binding energy differences in the interactions of CAT and other macromolecules

It should be noted that the CAT domain was defined to include the WGR structure to analyze the interaction between the WGR/CAT and DBD domains. The influence of ligands on the flexibility of the CAT structure was examined by analyzing the variation in RMSF. The influence of Lig1 on the structural flexibility of CAT is slight, as indicated by the slight difference in RMSF between the cases with and without Lig1 (**Fig 4A**). Lig2 can stabilize residue 200 to 300 (residue 730 to 830 in 4OPX), as implied by **Fig 4B**. On the contrary, when it comes to CAT monomer, Lig2 increases the flexibility of residue 100 to 200 (residue 630 to 730 in 4OPX), as shown in **S2A Fig in S1 File**. Lig3 slightly enhances the flexibility of residue 110 to 130 (the structure connecting WGR and HD; residue 640 to 660 in 4OPX) and residue 245 to 260 (the structure connecting HD and ART; residue 775 to 790 in 4OPX), as shown in **Fig 4C**. Similarly, the presence of Lig3 stabilizes the CAT monomer (**S2B Fig in S1 File**).

The binding mechanism of ligands binding with CAT was explored based on the energy decomposition of key residues at the interface where CAT interacts with other molecules. **Fig 4D** shows the interface diagram of CAT interacting with other molecules. Among them, the key residues of CAT on the interface of CAT/ZnF1 and CAT/ZnF3 belong to WGR and HD regions, which are marked in blue and yellow, respectively. The key residues of CAT interacting with DSB come from the WGR regions, and are depicted in red. In addition, the catalytic binding site marked in green in **Fig 4D** is far away from the interaction surface for CAT/DSB,

**Table 6. The calculated (MMGBSA) binding free energies of ZnF3/ZnF1.**

| Term | ZnF3/ZnF1 | ZnF3/ZnF1 ~Lig1 | ZnF3/ZnF1 ~Lig2 | ZnF3/ZnF1 ~Lig3 |
|---|---|---|---|---|
| $\Delta E_{vdw}$ | -77.93 | -78.18 | -69.58 | -96.51 |
| $\Delta E_{ele}$ | -517.84 | -591.05 | -316.85 | -656.56 |
| $\Delta E_{polar}$ | 572.58 | 633.15 | 385.91 | 734.71 |
| $\Delta E_{nonpolar}$ | -12.61 | -14.02 | -10.63 | -15.44 |
| $\Delta G$ | -35.81 | -50.10 | -11.15 | -33.81 |

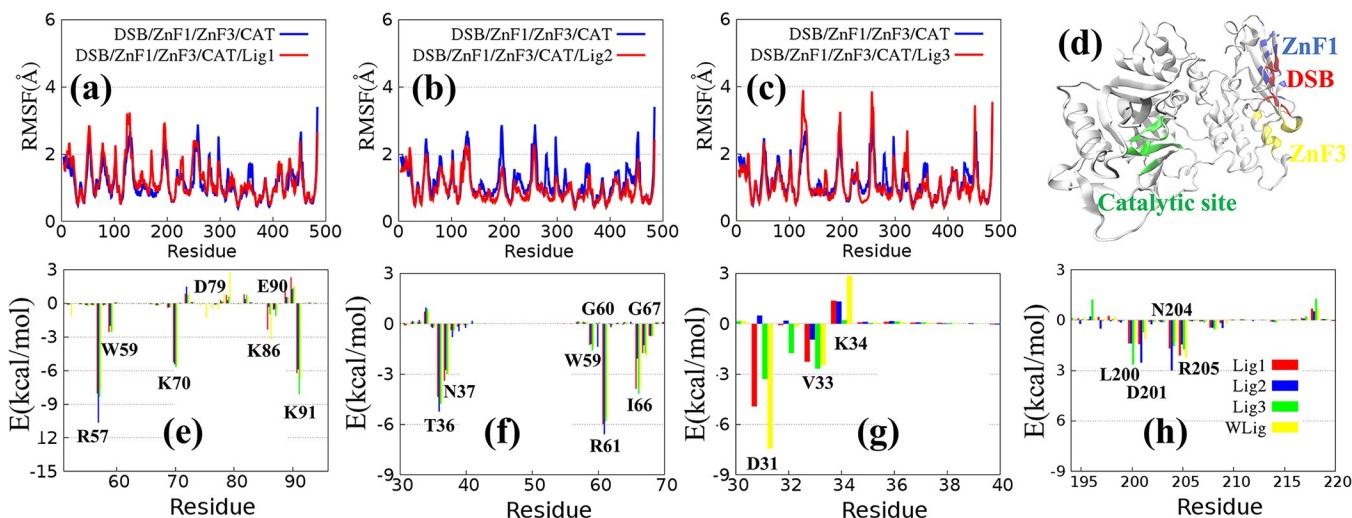

**Fig 4.** (a) Comparison of per-residue RMSFs between DSB/ZnF1/ZnF3/CAT and DSB/ZnF1/ZnF3/CAT/Lig1. (b) Comparison of per-residue RMSFs between DSB/ZnF1/ZnF3/CAT and DSB/ZnF1/ZnF3/CAT/Lig2. (c) Comparison of per-residue RMSFs between DSB/ZnF1/ZnF3/CAT and DSB/ZnF1/ZnF3/CAT/Lig3. (d) Three-dimensional structure diagram of CAT (4OPX). Interaction interfaces of CAT/DSB (residue 40 to 90), CAT/ZnF1 (residue 30 to 70), CAT/ZnF3 (residue 30 to 46 and residue 200 to 205) and catalytic site (residue 330 to 380) are represented by red, blue, yellow and green, respectively. (e) Per-residue energy contribution spectra of CAT on the surface of CAT/DSB. (f) Per-residue energy contribution spectra of CAT on the surface of CAT/ZnF1. (g-h) Per-residue energy contribution spectra of CAT on the surface of CAT/ZnF3. Finally, the color bars in (e-h) are the same, with red, blue, green, and yellow bars representing Lig1, Lig2, Lig3, and WLig (without ligand) binding to CAT, respectively. All current serial number of residues plus 530 are serial number of residues of CAT in 4OPX.

CAT/ZnF1 and CAT/ZnF3. **Table 7** shows the binding energy between CAT and DSB, as calculated by the MMGBSA method. The binding stability of CAT with DSB is weakened by Lig1, but strengthened by Lig2 and Lig3, which is mainly attributed to the variation in electrostatic energy and polar solvation free energy. Specifically, the sum of electrostatic energy and polar solvation free energy for the case with Lig1 is higher than its counterpart without Lig1, whereas the contrary is true for the cases of Lig2 and Lig3. Furthermore, the energy contribution of the key residues of CAT on the surfaces of CAT/DSB, CAT/ZnF1 and CAT/ZnF3 are displayed in (**Fig 4E–4H**) and **S3 Fig in S1 File**. The key residues affected by Lig1 are D79, E90, and L200. Lig2 mainly impacts W59, D79, G60, I66, D31 and D201. In contrast, Lig3 exerts a strong effect on K91, I66 and L200.

Finally, based on the analysis of **Tables 3, 5** and **7**, the effect of Lig1, Lig2, and Lig3 can be summarized as follows. Lig1 weakens the binding stability of CAT with ZnF1, ZnF3 and DSB, but strengthens the binding stability between ZnF1 and ZnF3. In contrast, Lig2 strengthens the binding stability of CAT with ZnF1 and DSB, but weakens the binding stability of CAT with ZnF3. In addition, Lig3 strengthens the binding stability of CAT with ZnF3 and DSB, but weakens the binding stability of CAT with ZnF1.

**Table 7. The calculated (MMGBSA) binding free energies of DSB/CAT.**

| Term | DSB/CAT | DSB/CAT~Lig1 | DSB/CAT~Lig2 | DSB/CAT~Lig3 |
|---|---|---|---|---|
| $\Delta E_{vdw}$ | -28.02 | -24.09 | -22.12 | -24.14 |
| $\Delta E_{ele}$ | -124.88 | -250.27 | -462.98 | -170.18 |
| $\Delta E_{polar}$ | 128.38 | 250.21 | 455.53 | 161.75 |
| $\Delta E_{nonpolar}$ | -4.52 | -3.85 | -3.38 | -3.68 |
| $\Delta G$ | -29.05 | -28.00 | -32.94 | -36.24 |

## Comparison of ligand–CAT interface for multi-domain and CAT monomer

Based on the influence of ligands on the interaction between CAT and other macromolecules, further analysis on the binding energy changes in CAT/ligand interactions was conducted. The binding energy between CAT and ligands can be calculated by two kinds of complexes. One is the combination of multiple domains and ligands, and the other is the combination of CAT monomer and ligands. The conformational changes in CAT in the monomer and the complex were compared between these two methods. (**Fig 5A–5C**) exhibits the differences in structural flexibility between CAT/DSB/ZnF1/ZnF3/ligand and CAT/ligand. For the combination of Lig2 and CAT, the interaction among CAT, DSB, ZnF1, and ZnF2 reduces the flexibility of CAT. On the contrary, for Lig1 and Lig3, the flexibility of CAT slightly increases with the binding of DSB, ZnF1 and ZnF3. **Fig 5D–5F** describes the per-residue energy contribution of CAT induced by the ligands. Among them, the repulsion of K363 and R348 in CAT

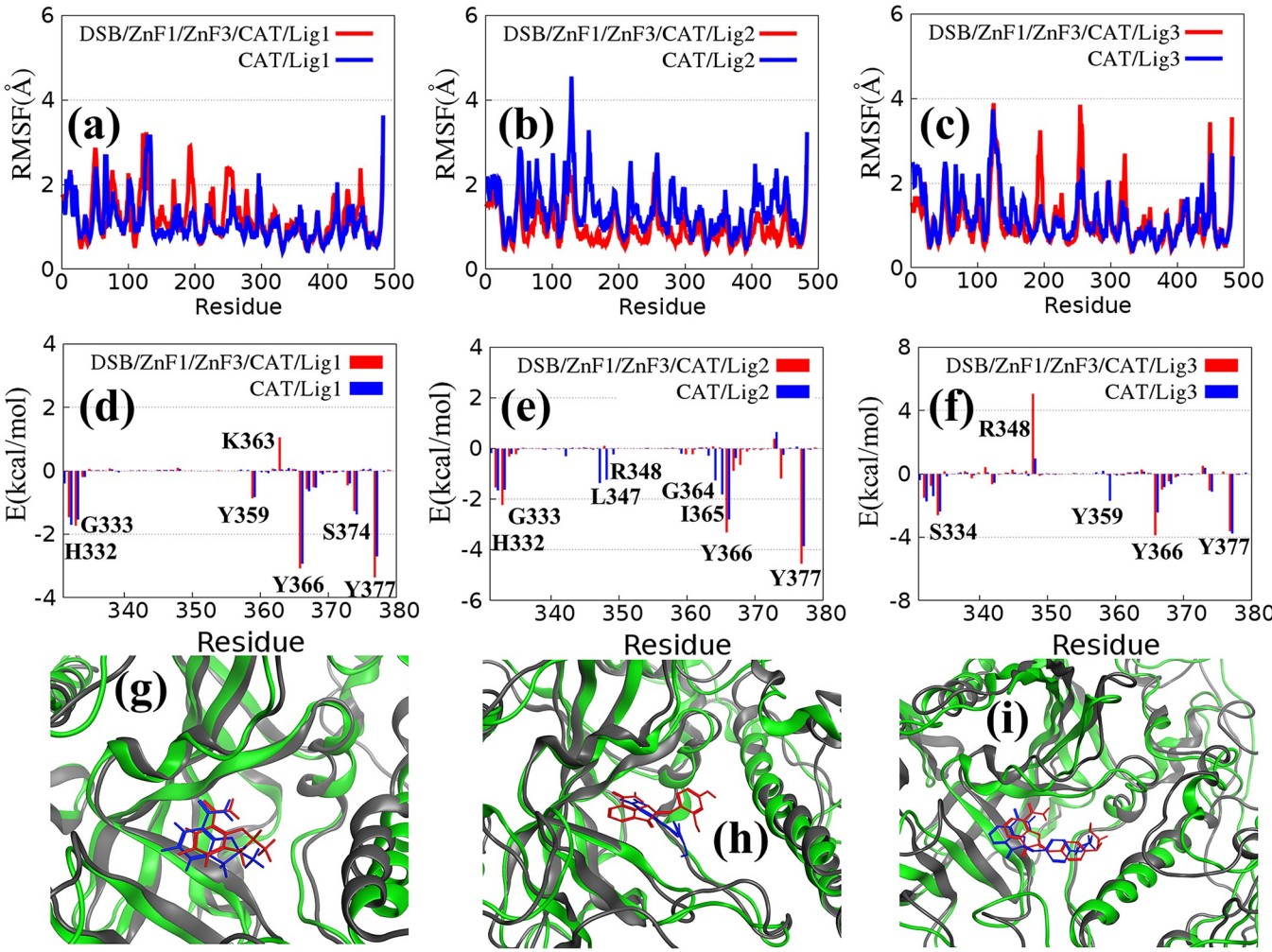

**Fig 5.** (a) Comparison of RMSFs for CAT between DSB/ZnF1/ZnF3/CAT/Lig1 and CAT/Lig1. (b) Comparison of RMSFs for CAT between DSB/ZnF1/ZnF3/CAT/Lig2 and CAT/Lig2. (c) Comparison of RMSFs for CAT between DSB/ZnF1/ZnF3/CAT/Lig3 and CAT/Lig3. (d-f) Per-residue energy contribution spectra of CAT on the surface of CAT/Lig1, CAT/Lig2 and CAT/Lig3, respectively. (g-i) Comparison of binding poses for these ligands between multi-domain and CAT monomers (4OPX, 4OQA and 4OQB). The complex containing the interactions between DSB, CAT, ZnF1, ZnF2 and ligands is depicted as a red stick model, and the blue one stands for the CAT and ligand complex.

significantly increases in the systems of DSB/ZnF1/ZnF3/CAT/Lig1 and DSB/ZnF1/ZnF3/CAT/Lig3. Furthermore, this repulsion is not obvious for Lig2. However, the interactions of key residues between DSB/ZnF1/ZnF3/CAT/Lig2 and CAT/Lig2, such as L347, R348, G364 and I365, are quite different. The main reason is the change in binding pose between the ligand and CAT domain induced by the interactions among CAT, DSB, ZnF1 and ZnF3 (Fig 5G–5I). The last sampling structure of the MD simulation trajectory was used to show the changes in the binding mode of the ligand. Compared with the system of DSB/ZnF1/ZnF3/CAT/ligand and CAT/ligand, the RMSD of Lig1 between DSB/ZnF1/ZnF3/CAT/Lig1 and CAT/Lig1 is 0.31 Å, which indicates a slight change in the binding mode of Lig1. The RMSD of Lig2 between DSB/ZnF1/ZnF3/CAT/Lig2 and CAT/Lig2 is 1.59 Å. This implies a large shift in the position and angle of Lig2, which brings about the change in key residue interactions (Fig 5H).

Finally, the RMSD of Lig3 between DSB/ZnF1/ZnF3/CAT/Lig3 and CAT/Lig3 is 0.49 Å, which indicates that the structure of Lig3 changes slightly, and the main change is a certain degree of movement in the horizontal direction. All results may be caused by some atoms of the non-nicotinamide group in the ligand. Lig1, which only contains the nicotinamide group, has a relatively stable binding mode with PARP1. For Lig2, the non-nicotinamide group has a strong polar effect which is easily affected by the protein pocket environment. Finally, for Lig3, the non-nicotinamide group is large with two ring groups with high structural stability. Therefore, the binding mode in different systems is only reflected in the difference in position.

The binding free energies of CAT/ligand and DSB/PARP1/ligand calculated by MMGBSA are reported in Table 8. For Lig2, the binding free energy of CAT/Lig2 is -24.15 kcal/mol, while in the system of DSB/PARP1/Lig2, the binding free energy increases significantly to -33.29 kcal/mol. This observation indicates that the use of CAT instead of PARP1 as the template for molecular docking may underestimate the binding energy between PARP1 and ligands. Therefore, unveiling the underlying mechanism behind this phenomenon may help to modify the structure of PARP1 inhibitors, thereby improving the structural stability of inhibitors based on the CAT structure as docking templates. Furthermore, considering the four subitems of binding free energy, all subitems are consistent with the total free energy in the PARP1/ligand systems. Moreover, these four subitems are also consistent with the total free energy in CAT/Lig1 and CAT/Lig3. However, for the system of CAT/Lig2, only the electrostatic energy $\Delta G_{ele}^{Lig2}$ between Lig2 and CAT is consistent with the total free energy, indicating that the electrostatic interaction between CAT and Lig2 is quite different from that between DSB/PARP1 and Lig2.

On the other hand, to compare the relationship between the binding energy and the binding affinity of these three ligands, we constructed a system (7kk6) for the interaction between veliparib and CAT where the three-dimensional structure of veliparib is shown in **S4A Fig in S1 File**, and the RMSD diagram of 100 ns MD simulation is recorded in **S4B Fig in S1 File**. In

**Table 8. The calculated (MMGBSA) binding free energies between ligands and CAT, or multi-domain of PARP1 with DSB (mPARP1).**

| Term | $\Delta E_{vdw}$ | $\Delta E_{ele}$ | $\Delta E_{polar}$ | $\Delta E_{nonpolar}$ | $\Delta G$ |
|---|---|---|---|---|---|
| CAT /Lig1 | -29.44 | -27.94 | 29.91 | -3.54 | -31.00 |
| CAT /Lig2 | -35.72 | -22.07 | 38.18 | -4.53 | -24.15 |
| CAT/Lig3 | -50.23 | -146.40 | 162.12 | -6.68 | -41.29 |
| mPARP1/Lig1 | -28.31 | -28.24 | 30.49 | -3.63 | -29.67 |
| mPARP1/Lig2 | -34.82 | -51.49 | 57.82 | -4.81 | -33.29 |
| mPARP1/Lig3 | -54.38 | -155.01 | 173.76 | -6.96 | -42.59 |
| CAT/veliparib | -37.10 | -130.62 | 137.72 | -4.04 | -34.03 |

addition, the protein flexibility of the CAT structure based on RMSF is shown in **S4C Fig in S1 File**. There are 352 amino acids in 7kk6, which does not contain the WGR domain. Consistent with Model 1 to Model 8, the last 20 ns MD simulation trajectory was used to calculate the binding energy between veliparib and CAT, as shown in **Table 8**. It shows that although the binding affinities of Lig1, Lig2 and Lig3 are weaker than that of veliparib, the binding free energies with PARP1 are similar to that of veliparib. Furthermore, the binding energies of these three ligands to PARP1 have a certain gradient suitable for studying the initial structure of the interaction between PARP1 and DSB regulated by ligands. However, the binding affinity of the ligands is not consistent with the binding free energy between the ligands and PARP1 or CAT. The main reasons are as follows. Firstly, multiple domains of PARP1 can be used as targets for a higher structural complexity. Secondly, there are several binding modes between ligand and PARP1. Finally, the method of calculating binding free energy can also bring errors. Therefore, it is difficult to evaluate ligand binding affinity by the binding free energy between ligands and CAT. However, the binding free energy reflects the binding stability of ligands with PARP1. Therefore, the relationship between binding free energy and the binding stability of ligands can be established. Meanwhile, this relationship is helpful to study the dynamic changes in the ligand in PARP1, and further helps to design and modify the ligand structure to obtain a new stable ligand structure with better binding affinity.

### Correlation analysis for CAT

To study the relationship of the interactions between CAT/ligands and CAT/DSB, linear correlation between all CAT residues was explored to describe the motion of CAT [24,34–37]. **Fig 6** shows the correlation of CAT residues in eight systems from Model 1 to Model 8, respectively. Areas with high correlation are marked by green triangles and rectangles. In this paper, the current sequence of CAT contains WGR, the HD sub-domain, and ART, including the catalytic site. It can be seen from **Fig 6** that the region contained in green triangles exhibits the

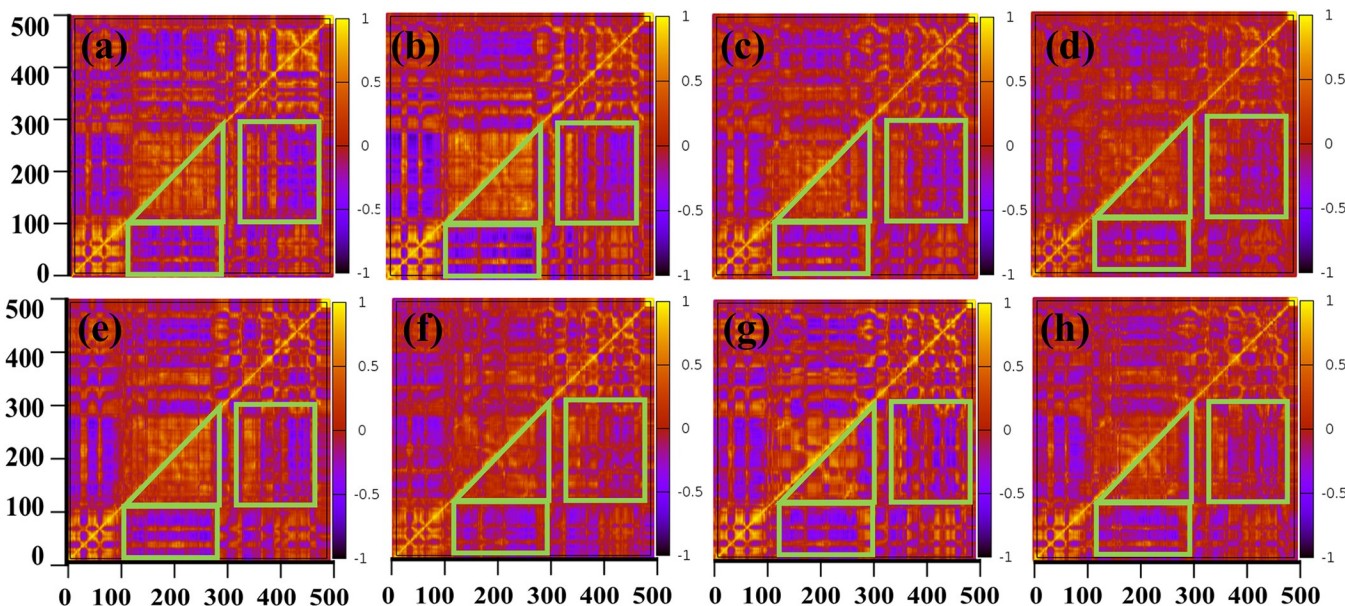

**Fig 6. Dynamic correlations of all residues in CAT.** (a) System of DSB/ZnF1/ZnF3/CAT. (b) DSB/ZnF1/ZnF3/CAT/Lig1. (c) DSB/ZnF1/ZnF3/CAT/Lig2. (d) DSB/ZnF1/ZnF3/CAT/Lig3. (e) CAT. (f) CAT/Lig1. (g) CAT/Lig2. (h) CAT/Lig3. The horizontal and vertical coordinates are both the sequences of CAT amino acid. The correlation ranges from 1 to -1, indicated by color changing from yellow to red and then blue.

correlation among residues within the HD sub-domain. There is a high positive correlation among these residues, indicating that the interactions between these residues are relatively strong, consistent with the stable structure of the helix. Among them, for the system combining DSB and PARP1, the combination of Lig1 indirectly increases the correlation within the HD domain, while for Lig2 and Lig3, the correlation of these regions shows a downward trend. A possible reason for this result is that the atoms of the non-nicotinamide group in the ligand can indirectly affect the correlation of HD by the interaction with multiple domains of PARP1 and DSB. Furthermore, for the system of the CAT monomer and the ligands, Lig2 promotes the correlation within the HD domain, while the other ligands decrease the correlation of these residues. This indicates that the changed binding mode of Lig2 leads to the phenomenon of inhibiting the internal correlation of HD, demonstrating the significance of using PARP1 with a complete structure to design and modify PARP1 inhibitors.

Possible transmission mechanisms deserve to be discussed. The RMSD of ZnF1/ZnF3/CAT/DSB between any two of the three systems (ZnF1/ZnF3/CAT/DSB/Lig1, ZnF1/ZnF3/CAT/DSB/Lig2, and ZnF1/ZnF3/CAT/DSB/Lig3) is below 0.1Å, which indicates that the ZnF1/ZnF3/CAT/DSB is nearly identical in the three systems if the interaction with ligands is not considered. Therefore, it is obvious that the observed negative correlation is induced by inhibitors. The catalytic binding site of CAT and the interaction interface of CAT, DBD and DSB are located at two different regions of CAT, far away from each other, as shown in **Fig 4D**. The two regions are connected by residues. The binding energies of ligand/CAT, ZnF1/CAT, and DSB/CAT are about -30kcal/mol, which implies that their interactions are stable. Consequently, it is inferred that the interaction between ligands and CAT can be transferred to the interaction between CAT and DSB by the negative correlation of domains in CAT.

In addition, it is also necessary to compare and clarify the differences in intra domain correlation between DSB/Znf1/Znf3/CAT and monomer CAT. Firstly, the apo-states of DSB/Znf1/Znf3/CAT and monomer CAT were compared. The correlation among WGR, HD and ART in these two systems shows no significant difference, which indicates that the interaction of CAT/Ligand rather than that of CAT/DSB and CAT/ZnF1 affects the intra domain correlation of CAT. Secondly, DSB/Znf1/Znf3/CAT/Ligand and CAT/Ligand were compared. The correlation of CAT between monomer CAT and complexes exhibits significant differences. For example, compared to the CAT/Lig1 system, Lig1 in DSB/Znf1/Znf3/CAT/Lig1 greatly enhances the negative correlation between WGR and HD, while Lig2 and Lig3 slightly weaken the positive correlation. To sum up, the ligand molecule directly interacts with the ART structure which contains the binding site. Meanwhile, the interaction is transferred to WGR through the structure of HD, thereby affecting the binding stability of WGR to the other domains of PARP1 and the molecule of DSB.

## Binding mechanisms among ligands, PARP1 and DSB

The relationships between ligands, PARP1 and DSB are further clarified in **Fig 7** based on above research on the interactions among ligands, CAT, ZnF1, ZnF3 and DSB structures. Firstly, among the three ligands, Lig1 has the highest binding free energy to CAT, while Lig3 has the lowest binding free energy to CAT (**Fig 7A** and **Table 8**). The interactions between ZnF1, ZnF3, and CAT were calculated to facilitate the analysis of the conformational changes in the DBD regulated by ligands. Lig2 enhances the interaction between CAT and ZnF1, while Lig1 and Lig3 weaken the binding stability between CAT and ZnF1 (**Fig 7B** and **Table 3**). Furthermore, Lig3 enhances the interaction between ZnF3 and CAT, while Lig1 and Lig2 exert a weakening effect (**Fig 7C** and **Table 5**). In addition, the interaction between ZnF1 and ZnF3 is strengthened by Lig1, but weakened by Lig2 and Lig3 (**Fig 7G** and **Table 6**). Therefore, ligands

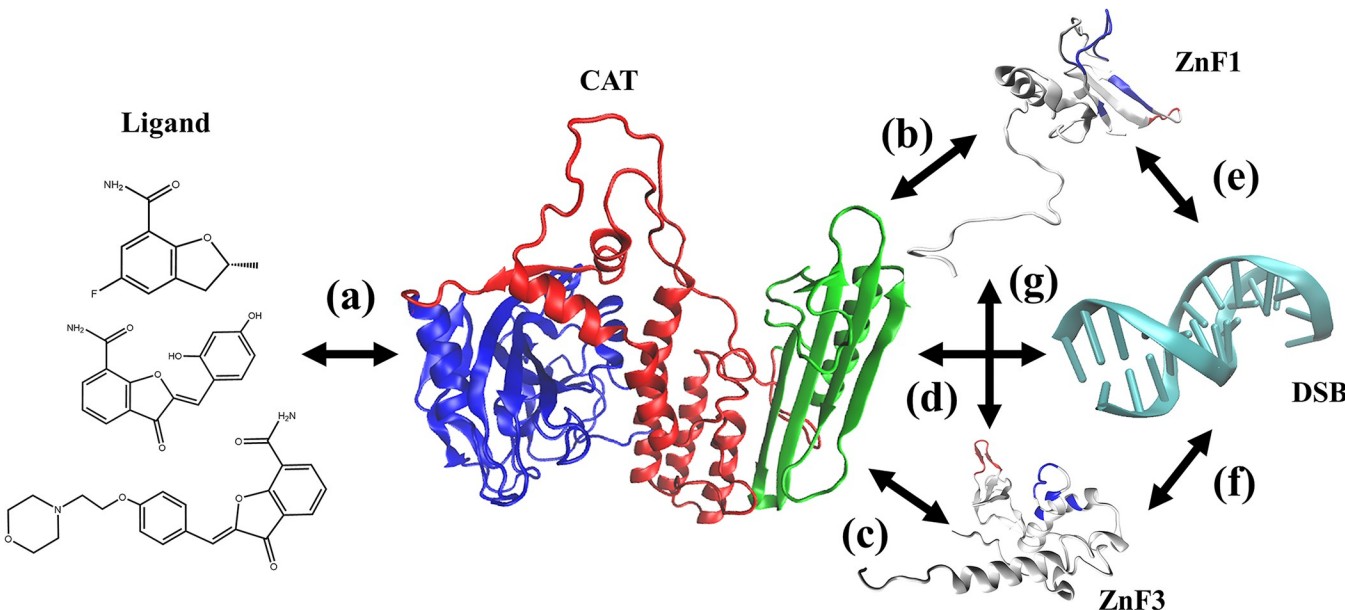

**Fig 7. Schematic diagram of binding mechanisms among ligands, CAT, ZnF1, ZnF3 and DSB (4OPX).** CAT area includes WGR (green cartoon), HD (red cartoon) and ART (blue cartoon). (a) Interaction between ligand and CAT. (b) Interaction between ZnF1 and CAT. (c) Interaction between ZnF3 and CAT. (d) Interaction between DSB and CAT. (e) Interaction between ZnF1 and DSB. (f) Interaction between ZnF3 and DSB. (g) Interaction between ZnF1 and ZnF3.

can regulate the binding stability among CAT and zinc-finger motifs in PARP1 through the interaction with CAT, and ultimately affect the binding stability of PARP1 with DNA.

Secondly, the three ligands have different effects on the binding stability of ZnF1 with DSB in relation to the function of PARP1. Specifically, Lig1 has the greatest weakening effect, followed by Lig2. In contrast, Lig3 has the slightest influence. In the binding system of DSB, PARP1 and the ligand, the stronger the binding stability of the ligand and CAT, the stronger the binding stability of ZnF1 to DSB (**Fig 7E**). This seems to contradict the activity of these ligands. In addition, the interaction between ZnF3 and DSB is weak which does not need to be considered first (**Fig 7F**). However, in the system of CAT monomers and ligands, the binding stability of Lig2 is the weakest, which is different with the system of DSB/CAT/ZnF1/ZnF3/ Lig2 (**Table 8**). Therefore, it is not reliable to use the binding energy between CAT and ligand as a measure of ligand affinity.

Finally, since PARP1 contains multiple domains, the complete structure of PARP1 has huge conformational space, with many possibilities for its functional conformation. For simplicity and efficiency, CAT monomer or CAT/Ligand are commonly used as the target for designing PARP1 inhibitors. Complete and accurate crystal structure of PARP1 is currently unavailable. Although computational methods can be used to predict the complete PARP1 structure, the predicted structure requires further structural optimization of MD simulation and experimental verification. The PARP1 structure used in this work contains most of the structural domains of PARP1. By introducing DSB and DBD domains, potential binding mechanisms for ligand regulation of PARP1 function were discovered. As shown in **Fig 7D** and **Table 7**, the enhancement in binding stability between ligands and CAT can strengthen the binding stability of CAT with DSB. According to the comparison with the binding free energy of ZnF1 and DSB, the stronger the binding stability between the ligand and CAT, the stronger the binding stability of PARP1 to bind with DSB. Therefore, using the complete

structure of PARP1 and DSB as a template for the prediction of inhibitor structure is beneficial for improving the accuracy of inhibitor.

## Conclusions

In the current study, detailed structural and dynamic conformational changes in multi-domains of PARP1 induced by ligands were investigated. Firstly, ligands slightly affect the structural flexibility of ZnF1 or ZnF3, but their influence on the binding stability between ZnF1/ZnF3 and DSB is relatively large, accompanied by changes in the energy contribution of some key residues. Secondly, ligands can indirectly influence the binding stability of CAT/DSB and ZnF1/DSB. Specifically, Lig2 enhances the binding stability between CAT and ZnF1. Lig3 enhances the interaction between CAT and ZnF3. Lig1 strengthens the interaction between ZnF1 and ZnF3. Thirdly, for multiple domains of PARP1, there is a strong negative correlation between the WGR and HD sub-domain in terms of structural changes, and HD and ART also show a strong negative correlation. In this manner, the interaction between ligands and catalytic binding sites can be transferred to the DNA recognition domain of PARP1 through the CAT domain, thus affecting the function of PARP1. Furthermore, it is not reliable to use the binding energy between the CAT and the ligand as a measure of ligand activity because it may differ greatly from the binding energy between the ligand and PARP1/DNA complex, such as in the case of Lig2. More importantly, the stronger the binding stability between ligand and CAT, the stronger the binding stability between PARP1 and DSB. PARP1 inhibitors can bind to catalytic binding site of PARP1 and thereby affect the motion correlation among the residues of CAT. Based on the correlation, PARP1 inhibitors indirectly affect the interaction between CAT and DNA recognition regions as well as DNA, ultimately inhibiting the DNA recognition and repair functions of PARP1. In the future, designing new ligands that can significantly affect the correlation of residues within CAT is an effective way to obtain new potential PARP1 inhibitors with better activity. In addition, docking potential inhibitor molecules with CAT/DSB/DBD is a good strategy to develop new potential PARP1inhibitors through computational methods. With the help of MD simulations, the binding energy of DBD and DNA can be calculated which can then be used to screen potential active inhibitor molecules. Therefore, this study can provide some useful guidance for the further design of PARP1 inhibitors.

## Supporting information

**S1 File. Contains supporting figures.**
(DOCX)

## Author Contributions

**Conceptualization:** Shuya Sun.

**Data curation:** Yizhou Wu.

**Funding acquisition:** Kai Wang.

**Investigation:** Shuya Sun.

**Methodology:** Lizhu Lai.

**Project administration:** Kai Wang.

**Validation:** Xin Wang.

**Writing – original draft:** Kai Wang.

**Writing – review & editing:** Shuya Sun.

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
