## [Decision Letter · Decision Letter 0]

12 May 2023

PONE-D-23-05051How ligands regulate the binding of PARP1 with DNA: deciphering the mechanism at the molecular levelPLOS ONE

Dear Dr. Wang,

Thank you for submitting your manuscript to PLOS ONE. After careful consideration, we feel that it has merit but does not fully meet PLOS ONE’s publication criteria as it currently stands. Therefore, we invite you to submit a revised version of the manuscript that addresses the points raised during the review process.

We look forward to receiving your revised manuscript.

Kind regards,

Amit Kumar

Academic Editor

PLOS ONE

Journal Requirements:

Reviewers' comments:

Reviewer's Responses to Questions

**Comments to the Author**

1. Is the manuscript technically sound, and do the data support the conclusions?

Reviewer #1: Partly

Reviewer #2: Yes

2. Has the statistical analysis been performed appropriately and rigorously? 

Reviewer #1: Yes

Reviewer #2: Yes

3. Have the authors made all data underlying the findings in their manuscript fully available?

Reviewer #1: Yes

Reviewer #2: Yes

4. Is the manuscript presented in an intelligible fashion and written in standard English?

Reviewer #1: No

Reviewer #2: Yes

5. Review Comments to the Author

Reviewer #1: The manuscript entitled “How ligands regulate the binding of PARP1 with DNA: deciphering the mechanism at the molecular level” by Wang et al, aimed to identify the mechanism and the effect of the inhibitor molecules on PARP1 binding to DNA by molecular dynamics simulation approach.

The work is not well-written. Some sentences need an improvement of the English language.

As example:

“To facilitate the development of potential inhibitors of PARP1 through structure-based computational design, it is of great significance to clarify the differences in structural dynamics and key residues between CAT/inhibitors and DNA/PARP1/inhibitors.” This sentence should be changed as “To facilitate the development of potential inhibitors of PARP1, it is of great significance to clarify the differences in structural dynamics and key residues between CAT/inhibitors and DNA/PARP1/inhibitors through structure-based computational design.”

Others concerns:

1.The Abstract mentioned “Furthermore, the interaction between ligands and catalytic binding sites can be transferred to the DNA recognition domain of PARP1 by a strong negative correlation movement among multi-domains of PARP1.”

But Figure 6 only explores the linear correlation between all CAT residues, that is, the interactions within the CAT residue domain (catalytic site domain). There is a strong negative correlation motion within CAT, and the combination of DSB and PARP1 causes changes in the CAT domain correlation. The conclusion should be more comprehensive. For example, the interaction between ligands and catalytic binding sites can affect DNA recognition domains, The combination of DSB and PARP1 also affects the conformation of CAT. Additionally, how can we determine if this negative correlation has transferred to the domain of DNA recognition? The author needs to clarify.

In addition to explaining the intra domain correlation between DSB/Znf1/Znf3/CAT and monomer CAT when combined with Lig1, Lig2, and Lig3, it is also necessary to compare and explain the differences in intra domain correlation between DSB/Znf1/Znf3/CAT and monomer CAT (longitudinal comparison).

Figure 6 shows that in the DSB/Znf1/Znf3/CAT system, Lig1 greatly enhances the negative correlation between WGR and HD, while Lig2 and Lig3 weaken the negative correlation. However, in the monomer CAT system, the binding of Lig1 does not have a significant or even reduced effect on the correlation; In monomer CAT, Lig2 enhances the negative correlation between HD and ART, while in the DSB/Znf1/Znf3/CAT system, Lig2 reduces the negative correlation and should be discussed separately.

2. “On the other hand, it is not reliable to use the binding energy between the CAT domain and the ligand as a measure of ligand activity, because it may in some cases differ greatly from the binding energy between the ligand and the PARP1/DNA complex. However, for PARP1/DNA/ligand, the stronger the binding stability between the ligand and PARP1, the stronger the binding stability between PARP1 and DNA.”

What is the significance of studying CAT/ligands? Isn't it enough to only study PARP1/DNA/ligands with practical significance?

3. The pictures are generally not clear enough.

The RMSF measurement of residual flexibility shown in Figure 2a of Figure 2 shows that the presence or absence of ligands can affect the flexibility of the ZnF1 structure, especially around the last 20 residues. The binding of ligands clearly increases the atomic fluctuations of the complex.

In Figure 2b, the interactions between Met38 and Phe39 and these structural domains should be depicted separately, and it is necessary to clarify how the residues on the interface between ZnF1 and DSB and CAT are determined.

Figures 2c and d are not clear enough. The article mentions the enhancement of A9 binding force, but they are not specifically labeled in Figures c and d. Specific amino acids need to be specifically labeled.

Similarly, what is the basis for determining the interface residues between ZnF3/DSB/CAT in Figure 3b? How are overlapping residues determined?

R345 is not indicated in Figure 5df.

Figure 5g-i should not be the last sampling structure of MD simulation track to display the change of ligand binding mode, but should use clustering method to find representative conformation to display the change of ligand binding mode.

Reviewer #2: The manuscript entitled “How ligands regulate the binding of PARP1 with DNA: deciphering the mechanism at the molecular level" by Kai Wang and coworkers explored the structural and dynamic conformational changes in multi-domains of PARP1 induced by ligands. The molecular dynamics calculations have been systematically performed and the results were analyzed extensively and correlated well with the energy decomposition method. Overall, this work sounds good and interesting to the readers.

However, the following points need to be addressed before publishing this manuscript.

1. Authors should provide the common name or class of the ligands and the IUPAC nomenclature of ligands.

2. The figure captions and figures are given in different places, it troubles a lot the reviewer to understand the results. It is highly inconvenient to critically analyze the results.

3. In all the figures, Kcal/mol is given wrongly instead of kcal/mol

4. In Figure 6, is there any label for the x, y, and Z axes? Are there any noticeable differences? What is the meaning of the different colors? It would have been given in the figure captions.

5. In conclusion, the authors should address, predicting the functional mechanism of PARP1 and the new directions to develop PARP1 inhibitors.

Other minor corrections:

1. In figure 4f, the Y axis label is partially hidden. It should be taken care of.

2. In materials and methods, in the MD simulation part, 1000 kcal/(mol•Å) - should be corrected.

3. In Table 2, Kcal/mol - should be corrected

4. In the whole manuscript, Kcal/mol – this correction should be carried out

6. PLOS authors have the option to publish the peer review history of their article (what does this mean?). If published, this will include your full peer review and any attached files.

Reviewer #1: No

Reviewer #2: **Yes: **Thirumoorthy Krishnan

---

## [Author Response · Author response to Decision Letter 0]

16 Jun 2023

Thank you a lot for your kind and careful reviewing. We greatly appreciate your valuable comments and suggestions, which helped us to improve the technical quality and presentation of our manuscript. All your comments have been addressed and the manuscript was revised accordingly. All modifications were highlighted in red in the revised manuscript. In addition, extensive English editing was carried out to improve the English expression of this paper. In the following, we give a point-by-point reply to your comments.

Reply to the reviewer 1

Comments to the Author

The manuscript entitled “How ligands regulate the binding of PARP1 with DNA: deciphering the mechanism at the molecular level” by Wang et al, aimed to identify the mechanism and the effect of the inhibitor molecules on PARP1 binding to DNA by molecular dynamics simulation approach.

The work is not well-written. Some sentences need an improvement of the English language.

As example:

“To facilitate the development of potential inhibitors of PARP1 through structure-based computational design, it is of great significance to clarify the differences in structural dynamics and key residues between CAT/inhibitors and DNA/PARP1/inhibitors.” This sentence should be changed as “To facilitate the development of potential inhibitors of PARP1, it is of great significance to clarify the differences in structural dynamics and key residues between CAT/inhibitors and DNA/PARP1/inhibitors through structure-based computational design.”

Response: We greatly appreciate the reviewer’s valuable comments and suggestions. Extensive English editing was carried out to improve the English expression of this manuscript. 

Others concerns:

1.The Abstract mentioned “Furthermore, the interaction between ligands and catalytic binding sites can be transferred to the DNA recognition domain of PARP1 by a strong negative correlation movement among multi-domains of PARP1.”

But Figure 6 only explores the linear correlation between all CAT residues, that is, the interactions within the CAT residue domain (catalytic site domain). There is a strong negative correlation motion within CAT, and the combination of DSB and PARP1 causes changes in the CAT domain correlation. The conclusion should be more comprehensive. For example, the interaction between ligands and catalytic binding sites can affect DNA recognition domains, The combination of DSB and PARP1 also affects the conformation of CAT. Additionally, how can we determine if this negative correlation has transferred to the domain of DNA recognition? The author needs to clarify.

Response: Thank you for pointing out this ambiguous description. We really appreciate it. Following your comment, we added some content to describe the correlation and interaction among ligands, CAT, ZnF1 and DSB more comprehensively in the revised manuscript. In addition, possible mechanisms for the negative correlation transmission were discussed in the revised manuscript. 

The content added to the Correlation analysis for CAT section of the revised manuscript is as follows:

“Possible transmission mechanisms deserve to be discussed. The RMSD of ZnF1/ZnF3/CAT/DSB between any two of the three systems (ZnF1/ZnF3/CAT/DSB/Lig1, ZnF1/ZnF3/CAT/DSB/Lig2, and ZnF1/ZnF3/CAT/DSB/Lig3) is below 0.1Å, which indicates that the ZnF1/ZnF3/CAT/DSB is nearly identical in the three systems if the interaction with ligands is not considered. Therefore, it is obvious that the observed negative correlation is induced by inhibitors. The catalytic binding site of CAT and the interaction interface of CAT, DBD and DSB are located at two different regions of CAT, far away from each other, as shown in Fig 4d. The two regions are connected by residues. The binding energies of ligand/CAT, ZnF1/CAT, and DSB/CAT are about -30kcal/mol, which implies that their interactions are stable. Consequently, it is inferred that the interaction between ligands and CAT can be transferred to the interaction between CAT and DSB by the negative correlation of domains in CAT.”

The content added to the Conclusions section of the revised manuscript is as follows:

“Secondly, ligands can indirectly influence the binding stability of CAT/DSB and ZnF1/DSB. Specifically, Lig2 enhances the binding stability between CAT and ZnF1. Lig3 enhances the interaction between CAT and ZnF3. Lig1 strengthens the interaction between ZnF1 and ZnF3.”

In addition to explaining the intra domain correlation between DSB/Znf1/Znf3/CAT and monomer CAT when combined with Lig1, Lig2, and Lig3, it is also necessary to compare and explain the differences in intra domain correlation between DSB/Znf1/Znf3/CAT and monomer CAT (longitudinal comparison).

Figure 6 shows that in the DSB/Znf1/Znf3/CAT system, Lig1 greatly enhances the negative correlation between WGR and HD, while Lig2 and Lig3 weaken the negative correlation. However, in the monomer CAT system, the binding of Lig1 does not have a significant or even reduced effect on the correlation; In monomer CAT, Lig2 enhances the negative correlation between HD and ART, while in the DSB/Znf1/Znf3/CAT system, Lig2 reduces the negative correlation and should be discussed separately. 

Response: Thanks for your valuable comments and suggestions. We really appreciate it. As suggested by the reviewer, we have added in-depth explanations and discussions to compare and explain the differences in intra domain correlation between DSB/Znf1/Znf3/CAT and monomer CAT. And meanwhile, some content was added to discuss the different mechanisms of ZnF1, ZnF3, and DSB on the binding process of CAT.

The content added to Correlation analysis for CAT section of the revised manuscript is as follows:

“In addition, it is also necessary to compare and clarify the differences in intra domain correlation between DSB/Znf1/Znf3/CAT and monomer CAT. Firstly, the apo-states of DSB/Znf1/Znf3/CAT and monomer CAT were compared. The correlation among WGR, HD and ART in these two systems shows no significant difference, which indicates that the interaction of CAT/Ligand rather than that of CAT/DSB and CAT/ZnF1 affects the intra domain correlation of CAT. Secondly, DSB/Znf1/Znf3/CAT/Ligand and CAT/Ligand were compared. The correlation of CAT between monomer CAT and complexes exhibits significant differences. For example, compared to the CAT/Lig1 system, Lig1 in DSB/Znf1/Znf3/CAT/Lig1 greatly enhances the negative correlation between WGR and HD, while Lig2 and Lig3 slightly weaken the positive correlation.”

The content added to the Conclusions section of the revised manuscript is as follows:

“PARP1 inhibitors can bind to catalytic binding site of PARP1 and thereby affect the motion correlation among the residues of CAT. Based on the correlation, PARP1 inhibitors indirectly affect the interaction between CAT and DNA recognition regions as well as DNA, ultimately inhibiting the DNA recognition and repair functions of PARP1.”

2. “On the other hand, it is not reliable to use the binding energy between the CAT domain and the ligand as a measure of ligand activity, because it may in some cases differ greatly from the binding energy between the ligand and the PARP1/DNA complex. However, for PARP1/DNA/ligand, the stronger the binding stability between the ligand and PARP1, the stronger the binding stability between PARP1 and DNA.”

What is the significance of studying CAT/ligands? Isn't it enough to only study PARP1/DNA/ligands with practical significance?

Response: Thank you for your valuable comment. We really appreciate it. Currently, it is impossible to only select PARP1/DNA/ligands for binding mechanism research and further inhibitor design. Firstly, since PARP1 contains multiple domains, the complete structure of PARP1 has huge conformational space, with many possibilities for its functional conformation. For simplicity and efficiency, CAT monomer or CAT/Ligand are commonly used as the target for designing PARP1 inhibitors. Complete and accurate crystal structure of PARP1 is currently unavailable. Although computational methods can be used to predict the complete PARP1 structure, the predicted structure requires further structural optimization of MD simulation and experimental verification. Therefore, predicting complete structure of PARP1 with structural optimization of MD simulation and developing new scoring function to compensate for the limitations of experimental methods in PARP1 crystallization can facilitate the development of potential PARP1 inhibitors. Finally, the PARP1 structure used in this work contains most of the structural domains of PARP1. And meanwhile, MD simulations and binding energy calculation were used to describe the binding mechanism among ligands, PARP1 and DSB. The strategy employed in this work is highly potential in obtaining reasonable PARP1 structure and useful for further research on molecular docking and inhibitor design. 

The content added to Binding mechanisms among ligands, PARP1 and DSB section is as follows:

“Finally, since PARP1 contains multiple domains, the complete structure of PARP1 has huge conformational space, with many possibilities for its functional conformation. 

For simplicity and efficiency, CAT monomer or CAT/Ligand are commonly used as the target for designing PARP1 inhibitors. Complete and accurate crystal structure of PARP1 is currently unavailable. Although computational methods can be used to predict the complete PARP1 structure, the predicted structure requires further structural optimization of MD simulation and experimental verification. The PARP1 structure used in this work contains most of the structural domains of PARP1. By introducing DSB and DBD domains, potential binding mechanisms for ligand regulation of PARP1 function were discovered. As shown in Fig 7d and Table 7, the enhancement in binding stability between ligands and CAT can strengthen the binding stability of CAT with DSB.”

3. The pictures are generally not clear enough.

The RMSF measurement of residual flexibility shown in Figure 2a of Figure 2 shows that the presence or absence of ligands can affect the flexibility of the ZnF1 structure, especially around the last 20 residues. The binding of ligands clearly increases the atomic fluctuations of the complex.

Response: Thank you for your careful reviewing. We really appreciate it. For the ZnF1 domain, the last 20 residues construct discrete loop structure which is far away from the center of ZnF1, as shown in the area protruding from the bottom left corner of Figure 2b. These residues of ZnF1 are far away from the functional interface of ZnF1/DSB and ZnF1/CAT. In addition, these residues not only have significant structural flexibility, but also have a certain degree of randomness. To ensure the completeness of the original crystal structure, we retained this structure in molecular dynamics simulation, but data analysis was not conducted on this flexible area. Therefore, the difference in structural flexibility in this area does not originate from ligands and does not need to be studied.

In Figure 2b, the interactions between Met38 and Phe39 and these structural domains should be depicted separately, and it is necessary to clarify how the residues on the interface between ZnF1 and DSB and CAT are determined.

Response: Thank you for your careful reviewing. We really appreciate it. Following your comment, we have modified the colors of Met38 and Phe39 in the revised manuscript to better show the interaction interface between ZnF1, DSB, and CAT. In addition, how to determine the residues on the interface between ZnF1, DSB, and CAT was explained in the revised manuscript.

The content added to Conformational changes in ZnF1 section is as follows:

“The interaction interface among ZnF1, DSB, and CAT was determined based on the distance between interacted atoms in the crystal structure of 4OPX. The specific steps are as follows: selecting a range of 4 angstroms among the atoms of residues or nucleotides, connecting these discontinuous residues together by other connected residues, and defining them as the interaction interface.”

Figures 2c and d are not clear enough. The article mentions the enhancement of A9 binding force, but they are not specifically labeled in Figures c and d. Specific amino acids need to be specifically labeled.

Response: Thank you for your careful reviewing. We really appreciate it. Following your comment, A9 was labeled in the latest Figure 2c and the key residues in Figure 2d were labeled accordingly. 

Similarly, what is the basis for determining the interface residues between ZnF3/DSB/CAT in Figure 3b? How are overlapping residues determined?

Response: Thank you for your careful reviewing. We really appreciate it. The basis for determining the interface residues between ZnF3/DSB/CAT in Figure 3b is similar to that for ZnF1/DSB/CAT. These overlapping residues determined by the residues in the defined overlap region depending on their proximity to other interacting residues or nucleotides.

R345 is not indicated in Figure 5df.

Response: Thank you very much for your careful reviewing. It should be R348. The mistake was corrected in the revised manuscript.

Figure 5g-i should not be the last sampling structure of MD simulation track to display the change of ligand binding mode, but should use clustering method to find representative conformation to display the change of ligand binding mode.

Response: Thank you for your suggestion. We really appreciate it. Clustering method is indeed one of the best methods for comparing macromolecular conformational changes. The method employed in this work conforms to the clustering method based on RMSD. The method is used to describe the change in the binding sites of the ligand structure which contains both translational and rotational components. It is found in this work that Lig 2 undergoes a certain degree of translation and rotation compared to the crystal structure, leading to changes in the key functional residues. What’s more, it is very convenient to use the clustering method based on RMSD for the purpose of structural comparison. As shown in Fig S1, the RMSD value is very stable at the late stage of MD simulation. Therefore, structures obtained from the late stage of MD simulation can be used to display the translational and rotational differences of ligand between stable MD simulation and crystal structure.

Reply to the reviewer 2

Comments to the Author

The manuscript entitled “How ligands regulate the binding of PARP1 with DNA: deciphering the mechanism at the molecular level" by Kai Wang and coworkers explored the structural and dynamic conformational changes in multi-domains of PARP1 induced by ligands. The molecular dynamics calculations have been systematically performed and the results were analyzed extensively and correlated well with the energy decomposition method. Overall, this work sounds good and interesting to the readers.

Response: We appreciate the reviewer’s valuable comments and suggestions.

However, the following points need to be addressed before publishing this manuscript :

1. Authors should provide the common name or class of the ligands and the IUPAC nomenclature of ligands. 

Response: Thank you for your suggestion. We really appreciate it. The names of Lig1, Lig2 and Lig3 were added in the revised manuscript.

The content added to the Materials and methods section of the revised manuscript is as follows:

“The names of Lig1, Lig2 and Lig3 are (2R)-5-fluoro-2-methyl-2,3-dihydro-1-benzofuran-7-carboxamide, (2Z)-2-(2,4-dihydroxybenzylidene)-3-oxo-2,3-dihydro-1-benzofuran-7-carboxamide, and (2Z)-2-{4-[2-(morpholin-4-yl)ethoxy]benzylidene}-3-oxo-2,3-dihydro-1-benzofuran-7-carboxamide, respectively.”

2. The figure captions and figures are given in different places, it troubles a lot the reviewer to understand the results. It is highly inconvenient to critically analyze the results.

Response: Thank you for your comment. The submission system has limitation on the number of words for each legend. Therefore, we had to separate legend from figures when we submitted the original manuscript. We apologize for the inconvenience caused. We will contact the editor to see if there is any way to solve this problem.

3. In all the figures, Kcal/mol is given wrongly instead of kcal/mol

Response: Thank you for pointing out this error. We have corrected this mistake in the revised manuscript.

4. In Figure 6, is there any label for the x, y, and Z axes? Are there any noticeable differences? What is the meaning of the different colors? It would have been given in the figure captions.

Response: Thank you for your comment. Following your comment, we revised the figure captions.

The content added to caption of Figure 6 is as follows:

“The horizontal and vertical coordinates are both the sequences of CAT amino acid. The correlation ranges from 1 to -1, indicated by color changing from yellow to red and then blue.”

5. In conclusion, the authors should address, predicting the functional mechanism of PARP1 and the new directions to develop PARP1 inhibitors

Response: Thank you for your value suggestion. Following your suggestion, we have added the potential functional mechanism and new directions to develop PARP1 inhibitors the Conclusions section of the revised manuscript.

The content added to the Conclusions section is as follows:

“PARP1 inhibitors can bind to catalytic binding site of PARP1 and thereby affect the motion correlation among the residues of CAT. Based on the correlation, PARP1 inhibitors indirectly affect the interaction between CAT and DNA recognition regions as well as DNA, ultimately inhibiting the DNA recognition and repair functions of PARP1. In the future, designing new ligands that can significantly affect the correlation of residues within CAT is an effective way to obtain new potential PARP1 inhibitors with better activity. In addition, docking potential inhibitor molecules with CAT/DSB/DBD is a good strategy to develop new potential PARP1inhibitors through computational methods. With the help of MD simulations, the binding energy of DBD and DNA can be calculated which can then be used to screen potential active inhibitor molecules.”

Other minor corrections:

1. In figure 4f, the Y axis label is partially hidden. It should be taken care of.

Response: Thank you for your comment. We really appreciate it. We have modified Fig 4f in the revised article.

2. In materials and methods, in the MD simulation part, 1000 kcal/(mol•Å) - should be corrected.

Response: Thank you for your careful reviewing. We really appreciate it. We have corrected it in the revised manuscript.

“1000 kcal/(mol•Å2)”

3. In Table 2, Kcal/mol - should be corrected. 

Response: Thank you for your careful reviewing. We really appreciate it. We have corrected it in the revised manuscript.

“All the energy terms are in kcal/mol.”

4. In the whole manuscript, Kcal/mol – this correction should be carried out.

Response: Thank you for your careful reviewing. We really appreciate it. We have corrected it in the revised manuscript.

---

## [Decision Letter · Decision Letter 1]

1 Aug 2023

PONE-D-23-05051R1How ligands regulate the binding of PARP1 with DNA: deciphering the mechanism at the molecular levelPLOS ONE

Dear Dr. Wang,

Thank you for submitting your manuscript to PLOS ONE. After careful consideration, we feel that it has merit but does not fully meet PLOS ONE’s publication criteria as it currently stands. Therefore, we invite you to submit a revised version of the manuscript that addresses the points raised during the review process. Please address the minor comments of Reviewer 1 and submit the paper. 

We look forward to receiving your revised manuscript.

Kind regards,

Amit Kumar

Academic Editor

PLOS ONE

Journal Requirements:

Reviewers' comments:

Reviewer's Responses to Questions

**Comments to the Author**

1. If the authors have adequately addressed your comments raised in a previous round of review and you feel that this manuscript is now acceptable for publication, you may indicate that here to bypass the “Comments to the Author” section, enter your conflict of interest statement in the “Confidential to Editor” section, and submit your "Accept" recommendation.

Reviewer #1: (No Response)

Reviewer #2: All comments have been addressed

2. Is the manuscript technically sound, and do the data support the conclusions?

Reviewer #1: Yes

Reviewer #2: Yes

3. Has the statistical analysis been performed appropriately and rigorously? 

Reviewer #1: Yes

Reviewer #2: Yes

4. Have the authors made all data underlying the findings in their manuscript fully available?

Reviewer #1: Yes

Reviewer #2: Yes

5. Is the manuscript presented in an intelligible fashion and written in standard English?

Reviewer #1: Yes

Reviewer #2: Yes

6. Review Comments to the Author

Reviewer #1: The manuscript entitled “How ligands regulate the binding of PARP1 with DNA: deciphering the mechanism at the molecular level” by Wang et al, aimed to identify the mechanism and the effect of the inhibitor molecules on PARP1 binding to DNA by molecular dynamics simulation approach.

The present study is logic and well designed, the analyses have been properly conducted, and the conclusions can be supported by their results. To further improve this study, there are some minor comments:

“Fig 5d-f describes the per-residue energy contribution of CAT induced by the ligands. Among them, the repulsion of K363 and R345 in CAT significantly increases in the systems of DSB/ZnF1/ZnF3/CAT/Lig1 and DSB/ZnF1/ZnF3/CAT/Lig3.” It should be R348. The mistake has not been corrected yet in the revised manuscript.

Reviewer #2: The revised version of the manuscript sounds good. No further revision is required. It can be accepted as such.

7. PLOS authors have the option to publish the peer review history of their article (what does this mean?). If published, this will include your full peer review and any attached files.

Reviewer #1: No

Reviewer #2: No

---

## [Author Response · Author response to Decision Letter 1]

1 Aug 2023

Thank you a lot for your kind and careful reviewing. We greatly appreciate your valuable comments and suggestions, which helped us to improve the technical quality and presentation of our manuscript. All modifications were highlighted in red in the revised manuscript. In the following, we give a point-by-point reply to your comments.

Reply to the reviewer #1

Comments to the Author

The manuscript entitled “How ligands regulate the binding of PARP1 with DNA: deciphering the mechanism at the molecular level” by Wang et al, aimed to identify the mechanism and the effect of the inhibitor molecules on PARP1 binding to DNA by molecular dynamics simulation approach.

The present study is logic and well designed, the analyses have been properly conducted, and the conclusions can be supported by their results. To further improve this study, there are some minor comments:

Response: We appreciate the reviewer’s valuable comments and suggestions.

“Fig 5d-f describes the per-residue energy contribution of CAT induced by the ligands. Among them, the repulsion of K363 and R345 in CAT significantly increases in the systems of DSB/ZnF1/ZnF3/CAT/Lig1 and DSB/ZnF1/ZnF3/CAT/Lig3.” It should be R348. The mistake has not been corrected yet in the revised manuscript.

Response: Thank you very much for your careful reviewing. The mistake was corrected in the revised manuscript.

“Among them, the repulsion of K363 and R348 in CAT significantly increases in the systems of DSB/ZnF1/ZnF3/CAT/Lig1 and DSB/ZnF1/ZnF3/CAT/Lig3.”

---

## [Editor Report · Decision Letter 2]

3 Aug 2023

How ligands regulate the binding of PARP1 with DNA: deciphering the mechanism at the molecular level

PONE-D-23-05051R2

Dear Dr. Wang,

We’re pleased to inform you that your manuscript has been judged scientifically suitable for publication and will be formally accepted for publication once it meets all outstanding technical requirements.

Kind regards,

Amit Kumar

Academic Editor

PLOS ONE
---

## [Editor Report · Acceptance letter]

7 Aug 2023

PONE-D-23-05051R2 

How ligands regulate the binding of PARP1 with DNA: deciphering the mechanism at the molecular level 

Dear Dr. Wang:

I'm pleased to inform you that your manuscript has been deemed suitable for publication in PLOS ONE. Congratulations! Your manuscript is now with our production department. 

Kind regards, 

on behalf of

Dr. Amit Kumar 

Academic Editor

PLOS ONE